# AttMetNet: Attention-Enhanced Deep Neural Network for Methane Plume Detection in Sentinel-2 Satellite Imagery

## Abstract

Methane is a powerful greenhouse gas that contributes significantly to global warming. Accurate detection of methane emissions is the key to taking timely action and minimizing their impact on climate change. We present AttMetNet, a novel attention-enhanced deep learning framework for methane plume detection with Sentinel-2 satellite imagery. The major challenge in developing a methane detection model is to accurately identify methane plumes from Sentinel-2's B11 and B12 bands while suppressing false positives caused by background variability and diverse land cover types. Traditional detection methods typically depend on the differences or ratios between these bands when comparing the scenes with and without plumes. However, these methods often require verification by a domain expert because they generate numerous false positives. Recent deep learning methods make some improvements using CNN-based architectures, but lack mechanisms to prioritize methane-specific features. AttMetNet introduces a methane-aware architecture that fuses the Normalized Difference Methane Index (NDMI) with an attention-enhanced U-Net. By jointly exploiting NDMI's plume-sensitive cues and attention-driven feature selection, AttMetNet selectively amplifies methane absorption features while suppressing background noise. This integration establishes a first-of-its-kind architecture tailored for robust methane plume detection in real satellite imagery. Additionally, we employ focal loss to address the severe class imbalance arising from both limited positive plume samples and sparse plume pixels within imagery. Furthermore, AttMetNet is trained on the real methane plume dataset, making it more robust to practical scenarios. Extensive experiments show that AttMetNet surpasses recent methods in methane plume detection with a lower false positive rate, better precision recall balance, and higher IoU.

## 1 Introduction

Methane is a potent greenhouse gas, accounting for approximately 20% of global warming since the industrial revolution Kirschke et al. (2013). Methane emissions originate from diverse anthropogenic sources, including agriculture, livestock, landfills, and the fossil fuel industry Saunois et al. (2019). Although methane has a relatively short atmospheric lifetime compared to carbon dioxide, it has a much higher global warming potential over a 20-year period US EPA (2016). Therefore, mitigating methane emissions offers a fast and effective strategy to slow climate change. Recent advancements in remote sensing technologies and multispectral imaging have enabled the identification of methane emission hotspots, facilitating timely and targeted mitigation efforts.

Methane exhibits strong absorption features in the shortwave infrared (SWIR) region of the electromagnetic spectrum, specifically between wavelengths of 1600–1850 nm and 2100–2500 nm Růžička et al. (2023). To capture data beyond the visible spectrum, several satellites equipped with multispectral sensors have been developed, enabling the detection of methane signatures in these SWIR bands. Among these, Sentinel-2 is a widely used satellite that captures imagery across 13 spectral bands spanning the visible, near-infrared (NIR), and SWIR regions. In particular, Bands 11 and 12 (B11 and B12) of Sentinel-2 cover the relevant SWIR ranges and can be utilized to extract methane ab-

sorption signals from observed scenes. In this paper, we propose *AttMetNet*, an attention-enhanced deep neural network designed to detect methane plumes using Sentinel-2 satellite imagery.

We formulate the methane detection task from two perspectives. Our model first determines whether an input image contains a methane plume and then generates a plume mask capturing the shape of the actual methane plume. Thus, our work involves both classification and segmentation tasks. The input to the model is a 12-channel raw Sentinel-2 image and the output is a single-channel plume mask. A scene is classified as containing a plume if the predicted mask's contiguous region of positive pixels exceeds a defined threshold.

However, the methane detection task in satellite imagery presents several significant challenges. First, the small and irregular shapes of methane plumes and the presence of noise in satellite images make the detection task complicated. Diverse land cover types and spectral overlap with surface artifacts (such as water vapor and $CO_2$) further increase complexity. Second, plume pixels can range from covering large areas to only a tiny fraction of satellite imagery, resulting in strong class imbalance. Moreover, the scarcity of labeled training data poses a major limitation, as methane plume events are rare and publicly available annotated datasets are limited. This leads to reliance on synthetic data that causes models not to generalize well to real-world scenarios.

To address these challenges, we propose AttMetNet, the first architecture that integrates the Normalized Difference Methane Index (NDMI) Webber & Kerekes (2020) with attention-based feature selection in a unified deep learning pipeline, significantly improving detection performance. The NDMI channel acts as a methane-sensitive input, guiding the network's attention toward plume regions and suppressing irrelevant background patterns. Simultaneously, attention gates dynamically prioritize spatial features relevant to plumes, enabling more precise localization of irregular methane emissions.

A key novelty of our framework lies in its integration of NDMI as an additional input channel along with the 12-band Sentinel-2 data. In remote sensing, spectral indices are mathematical functions of reflectance values at different wavelengths that enhance the detection of specific surface properties by highlighting features of interest and suppressing confounding factors. For methane detection, Sentinel-2's B12 band overlaps with the methane absorption region, while B11 band provides a nearby background reference. Therefore, subtracting B11 reflectance from B12 reflectance and normalizing the result creates NDMI, a spectral approximation of the presence of methane. Although NDMI alone is sensitive to contextual errors, we show that its incorporation into a deep learning pipeline for methane detection is novel and effective.

The second challenge is the severe class imbalance, as plume pixels often occupy a small proportion of satellite imagery. To minimize model bias, we employ focal loss, which assigns greater weight to hard-to-classify or rare examples while down-weighting well-classified ones. Despite the scarcity of real plume data, we intentionally rely on real-world data to train AttMetNet. We use the most up-to-date multispectral dataset of recorded methane plumes. This enables the model to generalize effectively to real satellite observations without reliance on synthetic augmentation, making it more robust and deployable in complex real-world scenarios.

Early works Varon et al. (2021); Ehret et al. (2022); Irakulis-Loitxate et al. (2022) were based on temporal differences and ratios between the B12 and B11 bands of Sentinel-2. These methods compare a scene with and without a methane plume to detect its presence. However, these methods remain highly sensitive to background context as methane absorption features can overlap with other gases (e.g., water vapor) or surface materials. This can cause numerous false positives that often require domain experts to verify for correct detections.

In recent works, deep learning methods have been explored to address these challenges. Most of the deep learning models Vaughan et al. (2024); Růžička et al. (2023); Rouet-Leduc et al. (2023) are based on the U-Net architecture and use different satellite datasets. U-Net treats all input data equally, which is good for general segmentation tasks, but it lacks mechanisms to prioritize methane-relevant features. Moreover, many works use synthetic datasets of simulated plumes due to the scarce real methane emission events Groshenry et al. (2022); Rouet-Leduc et al. (2023); Rouet-Leduc & Hulbert (2024). Models trained on such data may struggle to generalize actual methane emissions, leading to low reliability in practical applications.

**Our contributions are summarized as follows:**

- We present *AttMetNet*, the first methane plume detection framework that **jointly integrates NDMI with an attention-enhanced U-Net**, introducing a methane-aware design that selectively amplifies plume-relevant features while suppressing background noise.
- We systematically evaluate the impact of incorporating NDMI as an additional input channel and show that its integration sharpens the location of methane features and significantly improves the detection accuracy across models.
- We address the significant class imbalance inherent in methane plume segmentation through focal loss, demonstrating its effectiveness in handling both limited positive samples and sparse plume pixels to enhance model sensitivity to subtle methane emission patterns.
- We train and evaluate on a curated dataset of real methane plume events, providing evidence of model performance under real-world imaging conditions.

## 2 RELATED WORKS

Table 1: Comparison of related works on methane plume detection.

| Reference | Satellite | Sensor | Dataset | Model |
|---|---|---|---|---|
| Kumar et al. (2020) | AVIRIS-NG | Hyperspectral | Real | Mask-RCNN |
| Groshenry et al. (2022) | PRISMA | Hyperspectral | Synthetic | U-net |
| Rouet-Leduc et al. (2023) | Sentinel-2 | Multispectral | Synthetic | U-net |
| Kumar et al. (2023) | AVIRIS-NG | Hyperspectral | Real | Detection Transformer (DETR) |
| Růžička et al. (2023) | AVIRIS-NG | Hyperspectral | Real | U-net |
| Vaughan et al. (2024) | Sentinel-2 | Multispectral | Real | U-net |
| Rouet-Leduc & Hulbert (2024) | Sentinel-2 | Multispectral | Synthetic | U-net with ViT encoder |
| Si et al. (2024) | PRISMA, EnMAP | Hyperspectral | Synthetic | Mask-RCNN |
| **Our work (AttMetNet)** | **Sentinel-2** | **Multispectral** | **Real** | **U-net with attention gates** |

Methane detection from satellite imagery often uses physics-based and statistical approaches that rely on the absorption features of methane in specific infrared bands, such as Sentinel-2's B11 and B12. A widely used technique is the Multi-Band–Multi-Pass (MBMP) retrieval method Varon et al. (2021). This method works by comparing the B11 and B12 reflectance values of the same location taken at different times: one when a methane plume is present and another when it is not. By analyzing the reflectance value differences across two time-frames, the method can highlight areas where methane is likely present. However, MBMP depends on finding a suitable "clean" reference image without a plume, which can be difficult if emissions are persistent or frequent. The method can also produce errors due to ground surface or weather conditions which often requires domain experts to check the validity of methane plumes. Extensions like linear background projections of previous observations Ehret et al. (2022) or ratio-based approaches Irakulis-Loitxate et al. (2022) have been developed to address some of these challenges, but they still struggle in areas with rapidly changing landscapes or complex backgrounds. Matched filter methods Frankenberg et al. (2016); Duren et al. (2019); Cusworth et al. (2021); Thompson et al. (2016); Guanter et al. (2021); Irakulis-Loitxate et al. (2021) offer faster detection and directly estimate methane enhancements, but their performance drops in environments like water bodies or dark surfaces where background signals are more difficult to separate from methane plumes.

Recently, deep learning methods were used to surpass the results and shortcomings of traditional detection methods. Especially the CNN-based U-net architecture Ronneberger et al. (2015) has been widely applied in different research works. Groshenry et al. was the first to use U-net to generate methane concentration maps and plume masks of PRISMA satellite scenes Groshenry et al. (2022). Methanet proposed in Jongaramrungruang et al. (2022) used CNN for emission rate estimation of methane plumes. Joyce et al. developed a framework to generate both plume mask and emission rate from PRISMA satellite images Joyce et al. (2023). The models in the aforementioned works focus on hyperspectral satellite data and use synthetic plumes for training. Rouet-Leduc et al. employed a U-net model trained on simulated Gaussian plumes superimposed onto Sentinel-2 multispectral

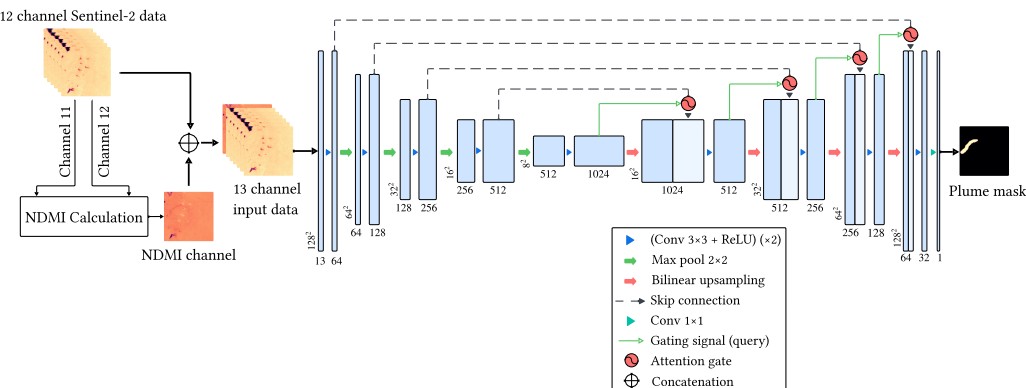

Figure 1: Architecture of AttMetNet. NDMI is first computed from channel 11 (B11 band) and channel 12 (B12 band) of raw 12-channel Sentinel-2 data. Then it is concatenated with the original 12 spectral channels to form a 13-channel input, which is fed into the Attention U-Net model.

imagery Rouet-Leduc et al. (2023). The usage of synthetic dataset limits generalization to real-world conditions due to the lack of true variability in the training data. To address this, Vaughan et al. (2024) introduces CH4Net, a U-net model trained on real Sentinel-2 plume events. But due to scarcity of recorded Sentinel-2 plume events, the dataset is significantly small. HyperSTARCOP and MultiSTARCOP models introduced in Růžička et al. (2023) leverage U-net architecture on AVIRIS-NG dataset consisting of hyperspectral images of real methane plumes. Some approaches using the Mask R-CNN model require extensive preprocessing of the input data to extract methane-specific features before segmentation can be performed Kumar et al. (2020); Si et al. (2024). Very few transformer-based models Kumar et al. (2023); Rouet-Leduc & Hulbert (2024) have also been explored, but their effectiveness is restricted by the scarcity of high-quality, annotated real plume data required for training.

Table 1 provides a concise summary of existing work on methane detection. In this work, we introduce the use of attention gates in the U-Net model for methane detection. It enables the model to focus more effectively on plume-related regions within a scene. Furthermore, utilizing real datasets enhances the model's robustness for real-world methane detection.

## 3 METHODOLOGY

AttMetNet is a methane-aware detection framework that integrates spectral-domain knowledge with attention-based feature learning. We introduce the Normalized Difference Methane Index (NDMI) as an additional input channel, enhancing methane-specific spectral signatures while suppressing background noise (see Section 3.1). The network processes the combined multispectral input with attention mechanisms that dynamically prioritize spatial features critical for localizing small, irregular methane plumes (see Section 3.2). To further improve sensitivity to sparse plume pixels, we incorporate focal loss, which effectively mitigates class imbalance and enhances detection of subtle methane emission patterns (see Section 3.3).

### 3.1 NORMALIZED DIFFERENCE METHANE INDEX

In remote sensing, spectral indices are commonly used for the detection of specific surface features. The Normalized Difference Vegetation Index (NDVI) utilizes red and near-infrared bands to assess vegetation health and biomass Rouse et al. (1973). Spectral indices are mathematical functions that take reflectance values at specific wavelengths as input and reduce the influence of unrelated surface elements such as soil, water, or uneven terrain, in order to highlight a particular feature of interest.

Methane shows high absorption in certain spectral bands, specifically in the Shortwave Infrared (SWIR) region, which can be leveraged to isolate plume regions from the background. Webber and Kerekes introduced the Normalized Difference Methane Index (NDMI), a spectral index designed

to enhance methane plume detection in multispectral satellite imagery Webber & Kerekes (2020). For Sentinel-2 bands, NDMI is calculated using B11 (1565 nm to 1655 nm) and B12 (2100 nm to 2280 nm). Methane exhibits higher absorption in B12 compared to B11, while the B11 band provides a background estimation due to its similar wavelength range. Subtracting B11 from B12 reduces background noise captured by the B11 band while preserving methane-specific features in the B12 band, thus amplifying the spectral signature of methane. NDMI is computed as follows:

$$\text{NDMI} = \frac{\text{B12} - \text{B11}}{\text{B12} + \text{B11}} \tag{1}$$

NDMI is computed for each pixel in the Sentinel-2 imagery and added as an additional channel to the original 12 spectral bands, resulting in a 13-channel input. While the NDMI channel is not immune to background artifacts, it provides a good approximation of methane features for a deep learning model to learn to distinguish methane plumes from background noise. Unlike the Multi-Band–Multi-Pass (MBMP) method Varon et al. (2021), which requires satellite images of the same scene acquired at different times to compare plume and non-plume conditions, NDMI offers a one-shot, computationally simple feature extraction. This helps the model focus more effectively on methane features without adding extra computational or data acquisition overhead.

## 3.2 Model Architecture

The core of our methane plume detection framework is built upon a U-Net architecture Ronneberger et al. (2015). To further enhance the model's ability to focus on methane plume regions, we integrate attention gates into the skip connections, inspired by the work of Oktay et al. Oktay et al. (2018). The framework is illustrated in Figure 1.

AttMetNet is a U-Net-based architecture designed to segment methane plumes from Sentinel-2 imagery. The input is a $128 \times 128$ image with 13 channels (12 spectral bands plus NDMI). The encoder consists of four convolutional blocks, each containing two sequences of $3 \times 3$ convolutions, ReLU activations, and batch normalization, followed by a $2 \times 2$ max-pooling layer. Each block doubles the number of feature maps ($64 \rightarrow 128 \rightarrow 256 \rightarrow 512$) while halving spatial dimensions ($128 \rightarrow 64 \rightarrow 32 \rightarrow 16$). The bottleneck block outputs 1024 feature maps at $8 \times 8$ resolution, capturing high-level semantics. The decoder reconstructs a high-resolution segmentation mask by progressively upsampling the encoded features ($1024 \rightarrow 512 \rightarrow 256 \rightarrow 128 \rightarrow 64$) using $2 \times 2$ transposed convolutions. At each stage, decoder features are fused with corresponding encoder features via skip connections enhanced by attention gates.

## 3.3 Loss Function

The attention mechanism selectively filters encoder features before fusion. Each attention gate takes the upsampled decoder features of the previous layer (the *gating signal*) and the corresponding encoder features through the skip connection as input. Both inputs are first re-

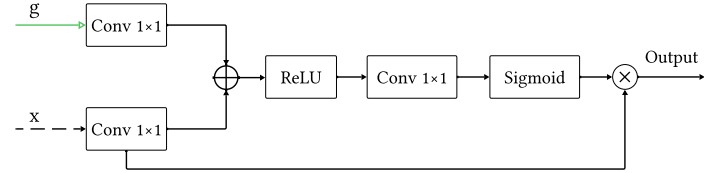

Figure 2: Structure of attention gate. Here, $g$ is the gating signal and $x$ is the encoder feature through the skip connection.

duced by $1 \times 1$ convolutions, combined through ReLU and another $1 \times 1$ convolution, and passed through a sigmoid to produce attention weights. These weights act as soft masks, highlighting plume-relevant regions while suppressing background noise.

The weighted encoder features are then fused with decoder features, followed by two $3 \times 3$ convolutions with ReLU and batch normalization. A final $1 \times 1$ convolution with sigmoid activation outputs the $128 \times 128$ binary mask (Figure 2).

After attention-based fusion, concatenated features are refined by two $3 \times 3$ convolutions with ReLU and batch normalization. The final $1 \times 1$ convolution with sigmoid activation generates the $128 \times 128$ binary mask.

Existing methane plume datasets present a high class imbalance, where positive plume instances are scarce compared to abundant non-plume samples, and plume pixels themselves occupy only small,

irregular regions within a scene. These factors make segmentation difficult when using standard cross-entropy loss, which treats all pixels equally.

To mitigate this, we employ *Focal Loss* Lin et al. (2018), which extends cross-entropy by introducing a modulating factor that reduces the contribution of well-classified examples and shifts focus toward harder ones:

$$\text{Focal Loss} = -\alpha_t(1 - p_t)^\gamma \log(p_t) \tag{2}$$

where $p_t$ is the predicted probability for the true class, $\alpha_t$ is a weighting factor to give more importance to the positive class, and $\gamma$ is a focusing parameter that prioritizes hard-to-classify examples.

By emphasizing difficult examples and reducing the influence of easy negatives, focal loss enhances the model's ability to detect subtle and irregular plume patterns, leading to improved segmentation performance in highly imbalanced scenarios.

## 4 EXPERIMENTS AND RESULTS

### 4.1 DATASET

Due to the limited availability of recorded methane plume events, real-world datasets are typically scarce and small in scale. To address this, we construct our dataset using the International Methane Emissions Observatory (IMEO)'s Eye on Methane platform[1] , which provides GeoJSON files containing plume metadata such as capture time, location (latitude and longitude), plume mask geometry, and other relevant information from six satellite sensors. From this resource, we select 1,656 plume events detected by the Sentinel-2 multispectral sensor. Using these metadata, we retrieve the corresponding L2A reflectance data via the Sentinel Hub API[2] , ensuring temporal and spatial alignment with the detected events. All spectral bands are resampled to 20 m resolution to match bands 11 and 12.

To complement the positive samples, we collect 4,458 negative samples (scenes without plumes), drawn both from regions near known plume sites and from unrelated locations. This design enables the dataset to support both targeted and generalized plume detection tasks. In total, the dataset consists of 6,114 images.

For training, each image is cropped into $128 \times 128$ pixel patches, corresponding to an area of approximately 6.55 km$^2$ at Sentinel-2's 20 m resolution. Random cropping is applied during training and validation to mitigate overfitting, while center cropping is used for testing. We further compute the Normalized Difference Methane Index (NDMI) from bands 11 and 12 and include it as a 13th input channel. To enhance robustness, we apply additional data augmentation techniques during training, including random rotations and Gaussian noise.

### 4.2 EXPERIMENT SETUP

***Baselines*** To evaluate our framework, we compare it against several baseline methods: **MBMP** Varon et al. (2021), **CH4Net** Vaughan et al. (2024), **U-Net with Convolutional Block Attention Module (CBAM)** Woo et al. (2018), **MultiResUnet** Ibtehaz & Rahman (2020), and **UNetFormer** Wang et al. (2022a). MBMP and CH4Net are specifically designed for methane detection, making them natural benchmarks for our task. CBAM has been successfully applied to various remote sensing tasks Li et al. (2020); Wang et al. (2022b); Cai et al. (2023), and we include it to evaluate its effectiveness in detecting methane. MultiResUnet is known for its ability to capture multiscale spatial features. It is included as baseline to assess whether enhanced spatial feature learning improves methane detection. Finally, UNetFormer, a UNet-like transformer architecture particularly designed for remote sensing segmentation, is included to examine the performance of data-intensive transformer models on limited multispectral datasets.

All baselines were initially trained on 12-channel Sentinel-2 data using standard binary cross-entropy loss. To evaluate the contributions of NDMI and focal loss, we also trained an extended

---

[1]https://methanedata.unep.org
[2]https://www.sentinel-hub.com

Table 2: Comparison of AttMetNet with baselines. The best performance is highlighted in bold.

| Method | Scene-level metrics | | | | | | | Pixel-level metrics | |
|---|---|---|---|---|---|---|---|---|---|
| | Accuracy | Balanced accuracy | Precision | Recall | F1 score | FPR | FNR | mIoU | Balanced accuracy |
| MBMP Varon et al. (2021) | 0.53 | 0.57 | 0.60 | 0.35 | 0.47 | 0.64 | 0.19 | 0.50 | 0.59 |
| CH4Net Vaughan et al. (2024) | 0.77 | 0.69 | **0.89** | 0.41 | 0.56 | 0.03 | 0.60 | 0.62 | 0.62 |
| CBAM U-net Woo et al. (2018) | 0.85 | 0.85 | 0.78 | 0.82 | 0.80 | 0.12 | 0.17 | 0.63 | 0.68 |
| CBAM U-net+ | 0.74 | 0.64 | 0.97 | 0.29 | 0.45 | **0.004** | 0.71 | 0.58 | 0.55 |
| MultiResUnet Ibtehaz & Rahman (2020) | 0.84 | 0.81 | 0.80 | 0.73 | 0.73 | 0.10 | 0.26 | 0.64 | 0.70 |
| MultiResUnet+ | 0.86 | 0.85 | 0.80 | 0.82 | 0.81 | 0.11 | 0.18 | 0.65 | 0.72 |
| UNetFormer Wang et al. (2022a) | 0.87 | 0.85 | 0.83 | 0.80 | 0.81 | 0.09 | 0.19 | 0.65 | 0.70 |
| UNetFormer+ | 0.83 | 0.80 | 0.83 | 0.67 | 0.74 | 0.07 | 0.32 | 0.61 | 0.66 |
| AttMetNet (Ours) | **0.89** | **0.88** | 0.83 | **0.86** | **0.85** | 0.09 | **0.12** | **0.66** | **0.75** |

Table 3: Performance comparison of AttMetNet with different loss functions.

| Loss function | Scene-level metrics | | | | | | | Pixel-level metrics | |
|---|---|---|---|---|---|---|---|---|---|
| | Accuracy | Balanced accuracy | Precision | Recall | F1 score | FPR | FNR | mIoU | Balanced accuracy |
| BCE loss | 0.81 | 0.74 | 0.87 | 0.54 | 0.67 | 0.04 | 0.46 | 0.66 | 0.67 |
| Weighted BCE loss | 0.87 | 0.85 | 0.84 | 0.80 | 0.82 | 0.07 | 0.20 | 0.65 | 0.73 |
| Focal loss | 0.89 | 0.88 | 0.83 | 0.86 | 0.85 | 0.09 | 0.12 | 0.66 | 0.75 |

version of the baselines (CBAM U-Net+, MultiResUnet+, UNetFormer+) using 13-channel input data, where NDMI is included as an additional channel, and focal loss replaces the BCE loss.

***Evaluation Metric*** To evaluate the performance of our methane plume segmentation model, we use two categories of metrics: **scene-level** and **pixel-level** Vaughan et al. (2024). Scene-level metrics evaluate the model's performance on the classification task, determining whether a methane plume is present in a scene. Pixel-level metrics assess the segmentation task, measuring how accurately the predicted plume mask matches the ground truth mask. These metrics together measure the model's ability to detect methane plumes and maintain a balance between foreground (plume) and background predictions. Below, we detail each category and its components.

- **Scene-level metrics**: An image is labeled as containing a plume if the predicted mask includes a contiguous region larger than 90 pixels. The 90 pixel threshold is based on the smallest plume size contained in the training dataset. Based on this binary decision, we evaluate the model using accuracy, balanced accuracy, precision, recall, false positive rate (FPR), and false negative rate (FNR).
- **Pixel-level metrics**: Predicted plume masks are compared to ground truth using metrics such as mean intersection over union (mIoU) and pixel-wise balanced accuracy.

***Training Configuration*** Our training set comprises 1,336 positive and 4,058 negative samples. For each epoch, we randomly sample twice the number of negative samples compared to positive samples, resulting in an effective training set of 1,336 positive and 2,672 negative samples. This approach helps prevent overfitting and ensures robustness to varying satellite terrain conditions. The test and validation sets contain 160 positive and 200 negative samples each, yielding an approximate 80–10–10 split for training, validation, and testing.

For training AttMetNet, we use a learning rate of 0.0001 while keeping other parameters at their default values. A plateau learning rate scheduler is employed with a decay factor of 0.5 and a patience of 7 epochs, and the model is trained for 100 epochs in total. Due to the severe class imbalance in our dataset, we use the focal loss function (Section 3.3). The positive class weighting factor $\alpha$ is set to 0.75, and the focusing parameter $\gamma$ is set to 2. These hyperparameters were determined experimentally.

## 4.3 COMPARISON WITH BASELINES

Table 2 presents a comparative analysis of our model's performance against baseline models across scene and pixel-level metrics. The traditional MBMP method underperforms due to its statistical design being prone to high signal-to-noise ratio and false positives, lacking adaptability to diverse land cover types. CH4Net, specifically designed for methane detection, achieves the highest precision and lowest false positive rate but suffers from extremely low recall, missing substantial methane emissions, which is problematic for comprehensive environmental monitoring.

Among deep learning baselines, CBAM U-Net shows balanced scene-level performance but struggles with pixel-level localization, while MultiResUnet exhibits solid pixel-level metrics but lower scene-level accuracy due to confusion between methane signatures and background patterns. UNet-Former achieves well-balanced performance with the second-highest F1 score but still lacks optimal precision-recall balance.

The enhanced baseline variants (+ models) reveal insights about architectural capacity under limited data conditions. MultiResUnet+ shows significant improvement when augmented with NDMI and focal loss, demonstrating effective utilization of our proposed components. However, CBAM U-Net+ and UNetFormer+ show minimal improvements, suggesting that with our limited training dataset of 1,336 positive samples, deeper architectures may struggle to effectively leverage additional spectral information due to overfitting or insufficient training data.

AttMetNet achieves the best overall performance with superior F1 score and mIoU while maintaining the lowest false negative rate. Our attention-enhanced architecture strikes an optimal balance between model complexity and data efficiency, effectively capturing methane signatures from limited real-world examples while ensuring computational efficiency suitable for practical satellite monitoring applications.

### 4.4 ABLATION STUDY

We ablate two key design choices in AttMetNet: (i) inclusion of NDMI as an input channel, and (ii) selection of the loss function for training under severe class imbalance.

#### 4.4.1 EFFECT OF NDMI (GRAD-CAM ANALYSIS)

To assess how NDMI guides spatial attention, we employ Grad-CAM Selvaraju et al. (2020) on two AttMetNet variants: 12-channel without NDMI and 13-channel with NDMI. Figure 3 shows that the NDMI-enabled model exhibits compact, localized activations aligned with ground-truth masks, while the 12-channel model produces diffuse activations extending beyond plume boundaries. This confirms NDMI provides methane-relevant cues that sharpen spatial focus and reduce false positives.

Figure 3: Comparison of Grad-CAM heatmaps illustrating AttMetNet activation with and without NDMI. Adding NDMI as a 13th channel results in more focused and accurate localization of target regions, as indicated by the closer correspondence between the heatmaps and ground truth.

#### 4.4.2 EFFECT OF FOCAL LOSS

Given extreme class imbalance in methane segmentation, we adopt Focal Loss Lin et al. (2018) which down-weights easy examples and concentrates learning on hard cases. Table 3 shows BCE yields high precision but low recall due to majority class bias. Weighted BCE improves precision-recall trade-off through class rebalancing. Focal loss achieves the best overall performance with highest balanced accuracy and F1 while maintaining strong recall, providing an effective solution for severe class imbalance.

### 4.5 CASE STUDIES

Figure 4 presents qualitative comparisons of segmentation results from our test set across diverse geographical regions including Turkmenistan, Algeria, USA, and Yemen. AttMetNet consistently produces masks closely aligned with ground truth in boundary delineation and spatial positioning across varying plume characteristics and environmental conditions. UNetFormer shows moderate performance but tends to produce fragmented masks, while MultiResUnet performs well on larger

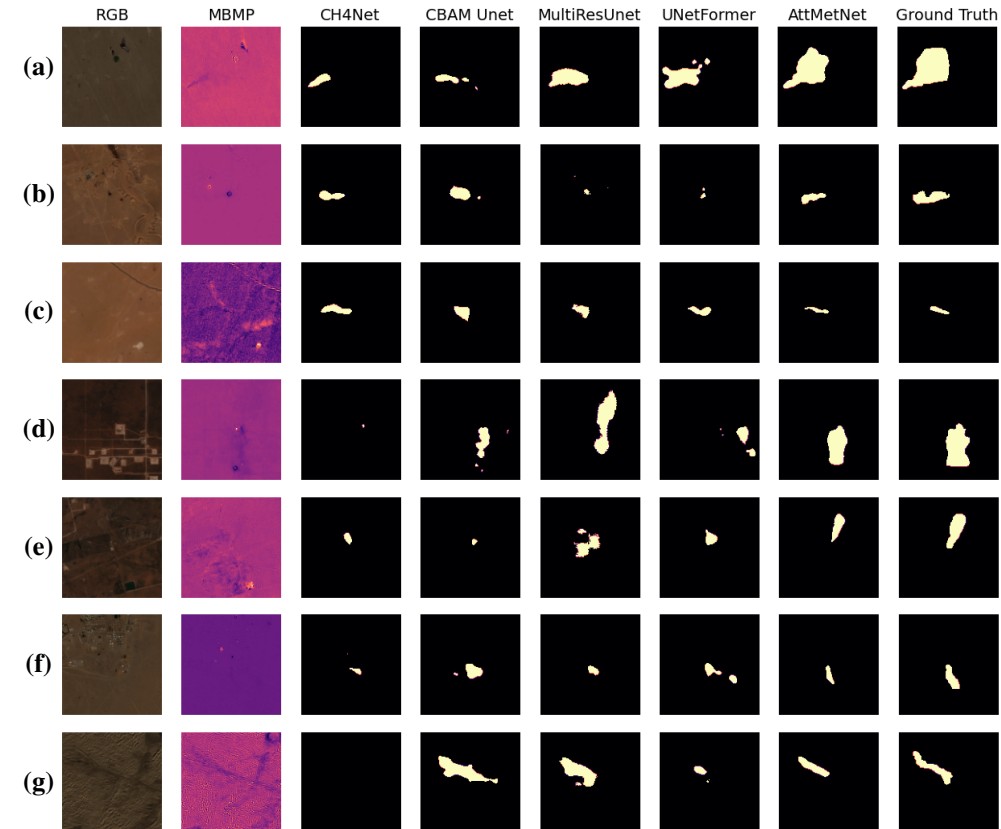

Figure 4: Comparison of different model predictions for methane plumes in different geographical regions. (a) Turkmenistan (38.5602°, 54.2129°) on 7 July 2024 (b) Algeria (28.6373°, 7.6165°) on 3 January 2024 (c) Algeria (31.7779°, 5.9951°) on 27 July 2023 (d) USA (32.1068°, -103.7154°) on 19 February 2024 (e) USA (32.3635°, -101.3277°) on 4 September 2023 (f) Yemen (15.5641°, 45.7987°) on 2 January 2023 (g) Turkmenistan (39.4614°, 53.7766°) on 1 December 2024.

plumes but struggles with smaller formations. CBAM U-net reliably detects plume presence but exhibits geometric inconsistencies, and CH4Net shows the most inconsistent performance, completely missing plumes in multiple scenes. These representative test samples demonstrate AttMetNet's superior consistency and accuracy across diverse geographical and atmospheric conditions.

## 5 CONCLUSION

We introduced AttMetNet, a novel deep learning framework for methane plume detection using Sentinel-2 satellite imagery. Built on a U-Net backbone enhanced with attention gates, AttMetNet focuses on methane-relevant spectral features while suppressing background artifacts. The integration of the Normalized Difference Methane Index (NDMI) as an auxiliary input channel further enhances attention to plume regions, while focal loss addresses the severe class imbalance inherent in methane plume segmentation where plume pixels constitute only a small fraction of total imagery.

Experiments on a real-world dataset from the International Methane Emissions Observatory (IMEO) demonstrate that AttMetNet outperforms both traditional methods and recent deep learning approaches across multiple metrics, including balanced accuracy, F1 score, and mIoU. The combination of attention mechanisms, NDMI integration, and focal loss effectively handles the challenging class imbalance while detecting methane plumes in complex environments.

Future work will extend this framework toward quantification and prediction of methane emissions, building upon the robust detection capabilities established in this study to enable comprehensive emission monitoring and assessment.

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
