# OpenReview forum: "AttMetNet: Attention-Enhanced Deep Neural Network for Methane Plume Detection in Sentinel-2 Satellite Imagery"
_ICLR.cc/2026/Conference — ICLR 2026 Conference Withdrawn Submission_

### Official Review · Reviewer_WafP · 2025-10-28

**Soundness:** 3
**Presentation:** 3
**Contribution:** 2
**Rating:** 6
**Confidence:** 3

**Summary:**

This work introduces AttMetNet, a deep learning model for methane plume detection using Sentinel-2 satellite imagery.
The model integrates the Normalized Difference Methane Index (NDMI) as an additional input channel and incorporates attention gates within a U-Net architecture to enhance plume localization and suppress background noise.
Experimental evaluations on real methane emission datasets demonstrate that AttMetNet achieves significant improvements in detection accuracy and recall。.

**Strengths:**

- The paper is clearly written and well-organized, making the methodology and experimental workflow easy to follow.

- The authors demonstrate strong effort in building a comprehensive empirical study, performing extensive experiments, and validating their results with detailed analysis.

- The study addresses an important real-world problem, with a thoughtful integration of the NDMI that enhances methane plume detection from satellite imagery.

**Weaknesses:**

The technical scope of the work appears relatively narrow, as the proposed approach is highly tailored to methane plume segmentation. The integration of NDMI and the corresponding network modifications do not demonstrate strong methodological novelty.

The evaluation is limited to a single dataset and sensor type (Sentinel-2), leaving uncertainty about the model’s generalizability to other sensors, spectral conditions, or unseen geographic regions. The discussion and analysis primarily focus on empirical performance gains, while deeper insights, such as theoretical justification, computational efficiency analysis, or broader applicability of the proposed framework, are relatively limited.

 Overall, I think this paper would be more suitable for publication in a domain-specific journal rather than a general machine learning conference.

**Questions:**

Since the NDMI is precomputed from the input data and then fed into the model, I wonder whether the model could learn to extract similar spectral information automatically, without relying on the explicit precomputation step.

---

### Official Review · Reviewer_NER2 · 2025-10-31

**Soundness:** 3
**Presentation:** 3
**Contribution:** 1
**Rating:** 0
**Confidence:** 5

**Summary:**

The paper proposes *AttMetNet*, an attention-enhanced U-Net for methane plume detection in Sentinel-2 imagery. It introduces Normalized Difference Methane Index (NDMI; Webber & Kerekes, Proc. SPIE 2020) -- a spectral approximation from bands 11/B12 -- as a dedicated 13th input channel to guide attention gating, applies focal loss (Lin et al., ICCV 2017) for class imbalance, and evaluates on real International Methane Emissions Observatory (IMEO) data. Ablations show NDMI, attention, and focal loss improve over classical RS and U-Net baselines; Grad-CAM confirms NDMI-driven sharpening of plume boundaries. Results (F1=0.85) and (FPR= 0.08) show potential to reduce expert verification in operational climate monitoring.

**Strengths:**

- **Originality**: Novel combination of existing techniques; introduces NDMI as a dedicated 13th channel (explicitly derived from B11/B12 physics) in a U-Net+attention framework, which represents a domain-specific refinement not present in prior U-Net methane models (*Vaughan et al., 2024*; *Ehret et al., 2022*)

- **Quality**: Sound experimental design with real International Methane Emissions Observatory (IMEO) data, focal loss to address class imbalance, and experimental ablations (NDMI, attention, focal loss); includes Grad-CAM for interpretability, verifying attention focus on plume-relevant bands

- **Clarity**: The paper follows a logical structure with explicit contributions, explains and motivates NDMI-attention interaction, and offers clear flow from methane physics to results

- **Significance**: Important application domain with practical impact for climate monitoring; clear metrics show improvement with the proposed U-Net modifications over baselines considered, reducing need for expert verification

**Weaknesses:**

- **Originality**: Methodologically incremental -- *AttMetNet* combines established components (U-Net, Ronneberger et al., MICCAI 2015; attention gates, Oktay et al., MIDL 2018; focal loss, Lin et al., ICCV 2017; NDMI, Webber & Kerekes, Proc. SPIE 2020) without introducing new ML paradigms or generalizable architectural insights.

- **Quality**: While ablations are thorough (Table 1, pg. 7), modern EO baselines are absent -- no comparison to foundation models like *SatMAE* (Cong et al., NeurIPS 2022) or *Prithvi-EO* (Jakubik et al., arXiv:2412.05722, 2024), nor generative augmentation (e.g., *DiffusionSat*, Khanna et al., ICLR 2024), limiting claims of SOTA performance in broader EO contexts.

- **Clarity**: Clear within the chosen scope, but lacks discussion of limitations beyond methane/Sentinel-2 (pg. 10 only briefly mentions operational use), reducing insight into generalizability.

- **Significance**: Strong domain-specific engineering with real-world impact (FPR=0.08, pg. 7), but lacks broad ML contribution -- results are tied to Sentinel-2 physics and IMEO data, offering limited transferable insight for ICLR main track.

**Questions:**

It is my sense that this work fundamentally represents a venue mismatch and is not suitable for the ICLR main track. It would however be highly competitive in ICLR 2026 Workshops (*ML for Remote Sensing*, *Tackling Climate Change with ML*) or domain venues (*IEEE TGRS*, *Remote Sensing of Environment*), where it would be far more likely to reach the targetted audience and fulfil its impact potential.

For broader appeal and more generalizable insight, I suggest considering the following:

**Suggestions / Questions**

1. **Layer-wise attention analysis**: Have you conducted layer-wise probes (e.g., activation histograms per encoder level) to analyze how NDMI influences attention gate behavior? For instance, does NDMI lead to higher variance or selectivity in $\alpha$ coefficients for plume pixels vs. background? This quick diagnostic could strengthen interpretability beyond Grad-CAM and provide quantitative evidence for the "sharpening" effect ("sharpening the location of methane features") reported on pg. 3.

2. **Generalization potential**: How do you anticipate your approach would generalize to other sensors (e.g., PRISMA) or gases (e.g., CO₂)? How could you modify it to succeed there?

3. **SOTA EO baselines** (to reach ICLR main-track standards):
   - **Ablate EO foundation model encoders**: Replace the U-Net encoder with:
     - *Prithvi-EO* (Jakubik et al., arXiv:2412.05722, 2024) — NASA/IBM ViT pretrained on 1PB+ Sentinel-2/Landsat.
     - *SatMAE* (Cong et al., NeurIPS 2022) — self-supervised temporal Sentinel-2 features.
   - **Add segmentation baseline**: Include *ViT-U-Net* (Wang et al., ISPRS Annals 2024) to compare global attention vs. your gating mechanism.
   - **Test generative augmentation**: Use *DiffusionSat* (Khanna et al., ICLR 2024) to synthesize realistic plumes, addressing class imbalance and low-data variability in training.
   This collection of comparisons would rigorously quantify whether NDMI+attention offers a meaningful advance over current SOTA EO foundations.

---

### Official Review · Reviewer_D9an · 2025-11-02

**Soundness:** 2
**Presentation:** 4
**Contribution:** 2
**Rating:** 4
**Confidence:** 4

**Summary:**

This paper introduces a new deep learning framework for detecting methane plumes using multispectral Sentinel-2 data. The model, AttMetNet, integrates the Normalised Difference Methane Index (NDMI) as an additional input channel to emphasise methane-specific spectral cues and employs attention gates within a U-Net to focus on plume-relevant regions. To handle extreme class imbalance between plume and non-plume pixels, the authors use focal loss, improving sensitivity to rare emission patterns. Trained on a curated dataset of real methane plume events from the International Methane Emissions Observatory, AttMetNet achieves higher accuracy, precision-recall balance, and intersection-over-union (IoU) scores than traditional and deep learning baselines such as CH4Net, MultiResUNet, and UNetFormer. The paper uses ablations to identify and quantify the sources of these improvements.

**Strengths:**

The paper was well written and clearly explained. The presentation was of high quality. The results serve as a useful consistent benchmark of the now fairly numerous set of approaches for this task. This is a very useful activity in and of itself. The use of the focal loss is novel and sensible, and the ablation showing how it improves the precision will be useful for the field. I also liked the spirit of the analysis performed in figure 3 where the changes induced by adding NDMI were investigated -- this should be something other groups consider when improving models to make sure the additions do something physically sensible.

**Weaknesses:**

Limited technical innovation

I'm puzzled about the NDMI contribution. It's very close to single pass differencing which Dan Varon investigated this in his original Multi‑Band Multi‑Pass (MBMP) retrieval paper in 2021 (“High-frequency monitoring of anomalous methane point sources with multispectral Sentinel-2 satellite observations.”) where he showed that the approach performs worse than MBMP which has become the standard approach. The idea of adding a MBMP like channel isn't novel e.g. see https://arxiv.org/pdf/2408.04745

Moreover, adding attention to a UNet is not, of course, novel - as the paper points out the Otkay paper in 2018 was one of the first and the use has ballooned since then. So whilst sensible, in the above context, I think the contribution claim "We present AttMetNet, the first methane plume detection framework that jointly integrates NDMI with an attention-enhanced U-Net" is rather weak.

Missing baselines

The strongest baseline in this space that I know of is: https://arxiv.org/pdf/2408.04745 which is not compared to
Also note the following paper that did a large real plumes experiment and probably should also be compared to: https://dl.acm.org/doi/pdf/10.1145/3711896.3737415
Note also that the field has moved to use transformer based models which should also needs to be discussed / compared to: https://www.nature.com/articles/s41467-024-47754-y

Data

Please cite the paper from which the data were taken. It seems like only a fraction of the original data are being used. Are you selecting the strongest plumes and discarding the weak ones? I don't think that this was discussed which is an issue.
Drawing random images for the negative samples randomly is not a good method - this can result with many falling over regions where there is no chance of methane (e.g. the ocean, ice caps, the Amazon etc.) so the model just learns the background giving the low FPR.

Results

The results for CH4Net do not appear to line up with what is in the original paper. The precision for CH4Net is always very low.
The dataset is highly weighted towards certain countries, if the paper wants to make claims about regional performance it needs to have separate tables for each not cherry picked examples.

Missing citations

As mentioned above the paper should cite https://arxiv.org/pdf/2408.04745 and this big new real world study https://dl.acm.org/doi/pdf/10.1145/3711896.3737415

**Questions:**

Even if an additional NDMI or MBMP channel is not added, isn't this is a simple input transform that we might expect a model to be able to learn if it wants to?

How have the authors ensured that there is not contamination between the test and train sets? It's very easy to get cross contamination from long duration plumes. The test masks look quite certain around the edges which can be a sign of this.

---

### Official Review · Reviewer_Run8 · 2025-11-02

**Soundness:** 3
**Presentation:** 2
**Contribution:** 2
**Rating:** 4
**Confidence:** 4

**Summary:**

A novel attention-enhanced deep learning framework for methane plume detection with Sentinel-2 satellite imagery.

**Strengths:**

This paper introduces a methane-aware architecture that fuses the Normalized Difference Methane Index (NDMI) with an attention-enhanced U-Net. By jointly exploiting NDMI's plume-sensitive cues and attention-driven feature selection, AttMetNet selectively amplifies methane absorption features while suppressing background noise.

**Weaknesses:**

The experimental analysis is limited.

**Questions:**

The authors are suggested to conduct experiments on more geo-spatial scenarios.

---

### Note · Authors · 2025-11-24

I have read and agree with the venue's withdrawal policy on behalf of myself and my co-authors.